# Structure-Aware Consistency Priors for Shape from Polarization in Complex Media

**Kaimin Yu** [1]  **Puyun Wang** [1]  **Huayang He** [2]  **Xianyu Wu** [† 1]

## Abstract

Recovering surface normals from single-view polarization images in complex media remains challenging. This paper focuses on ice as a representative complex medium, where intricate light–matter interactions lead to a nonlinear mapping between polarization observations and surface normals. To address this, a structure-aware polarization prior based on autocorrelation functions is proposed to capture the local spatial consistency of AoLP. Building on this, a dual-branch network (IceSfP) is designed to integrate raw polarization features with priors via cross-modal attention and multi-scale feature fusion, enabling accurate surface normal estimation under complex media conditions. To evaluate the method, the first real-world ice SfP dataset is constructed. Experimental results show that the method outperforms existing approaches across all metrics, achieving a MAE of $16.01°$, which is $2.74°$ lower than the second-best method. The framework provides a generalizable solution for high-precision geometric perception in complex media.

## 1. Introduction

Recovering surface normals from single-view observations is a fundamental problem in geometric perception, with significant implications in computer vision and machine learning (Wang et al., 2025a; Yang et al., 2026; Wang et al., 2025b; Tang et al., 2025). Considerable progress has been made under ideal reflection models and simplified imaging assumptions. However, in real-world media with complex material properties and internal light transport mechanisms, classical assumptions often fail, resulting in highly nonlinear and unstable relationships between observed signals and

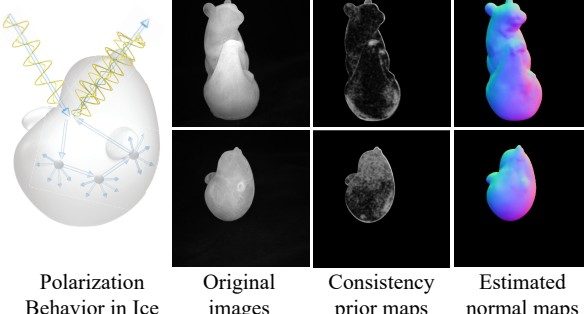

Polarization   Original   Consistency   Estimated
Behavior in Ice   images   prior maps   normal maps

*Figure 1.* Light behavior in ice and single-view surface normal estimation. Unpolarized incident light undergoes multiple internal scattering events within ice, leading to severe perturbations of polarization states and producing emergent light dominated by mixed surface and volumetric components. Original polarization image, structure-aware polarization consistency prior, and estimated surface normals are shown.

true surface geometry (Wang et al., 2025c; Zhang et al., 2025; Braun et al., 2024; Feng et al., 2025). Recovering accurate surface geometry under such conditions thus remains a significant challenge (Wang et al., 2025d; Marlow & Anderson, 2024).

Ice media provide a concrete and practically important scenario for studying geometric perception under such challenging observation conditions. When incident light interacts with ice, only a fraction is directly observed through surface reflection, while a significant portion undergoes complex internal processes, including birefringence, multiple scattering, and anisotropic propagation (Xu et al., 2023; Ziyu et al., 2024), as illustrated in Fig. 2. These interactions lead to substantial spatial reconfiguration of both polarization state and light paths, causing highly unstable correspondences between pixel-level observations and true surface normals. Meanwhile, accurate characterization of ice geometry directly affects mechanical, thermal, and aerodynamic responses, and is critical for applications such as environmental monitoring, road icing detection, the food industry, and polar research (Müller et al., 2024; Zhang et al., 2024). Motivated by these challenges and applications, this work focuses on single-view surface normal estimation for ice objects.

Due to the highly complex and unstable optical properties of

---

[1]The School of Mechanical Engineering and Automation, Fuzhou University, Fuzhou, China [2]Research Institute of Highway, Ministry of Transport, Beijing, China. Correspondence to: Xianyu Wu[†] <xwu@fzu.edu.cn>.

*Proceedings of the 43$^{rd}$ International Conference on Machine Learning*, Seoul, South Korea. PMLR 306, 2026. Copyright 2026 by the author(s).

ice, existing 3D imaging methods often struggle to achieve reliable performance in such scenarios (Chen et al., 2025). For active imaging, intricate refraction paths and strong scattering effects make it difficult to explicitly model the correspondence between projected signals and true surface geometry, rendering the inversion process highly ill-posed. Passive imaging methods, in contrast, rely on surface texture and disparity correspondences, which are unreliable on ice surfaces, leading to unstable matches and highly uncertain geometric inference. Although prior studies have attempted to address these challenges by introducing additional sensing modalities or sophisticated hardware configurations, such solutions typically require expensive equipment and controlled imaging conditions, significantly limiting their scalability and practical applicability (Zuo et al., 2024; Gou et al., 2023; Cai et al., 2026).

Shape from Polarization (SfP) provides a passive, low-cost alternative for geometric shape estimation while offering physically interpretable surface constraints (Zhu et al., 2025; Li et al., 2023; Cai et al., 2023). By modeling the polarization measurements of reflected light, SfP constrains surface normals through the Fresnel reflection model. Traditional SfP methods often rely on analytic models, but their performance degrades significantly in complex optical media, where multiple scattering, birefringence, and anisotropic propagation cause observed signals to deviate from idealized models. Recent deep learning-based approaches have shown promise in capturing the complex mapping between polarization observations and surface geometry, partially alleviating the limitations of incomplete physical models (Ba et al., 2020; Peng et al., 2025; Li et al., 2025a;b). However, when polarization signals are disturbed by complex optical effects and exhibit significant spatial inconsistencies, achieving robust single-view geometric reconstruction remains highly challenging.

To address these challenges, we propose a polarization consistency prior constructed using the autocorrelation function to characterize the local spatial coherence of polarization observations. Building on this, we design a dual-branch network architecture that adaptively leverages polarization-derived normal priors within a learning framework. The introduction of consistency priors improves the accuracy of single-view surface normal estimation while endowing the network with enhanced physical interpretability, effectively integrating physics-based imaging models with the deep learning paradigm. To validate the effectiveness of our method, we further construct the first real-world ice object SfP dataset.

The main contributions of this work are summarized as follows:

- A single-view polarization-based surface normal recovery method tailored for ice media is proposed.

- A structure-aware polarization consistency prior is proposed, constructed from the angle of linear polarization (AoLP) autocorrelations, and combined with a Cross-modal Reliability Attention (CRA) module to selectively weight physics-based normal priors, enabling accurate surface normal estimation under ambiguous polarization signals.

- The first real-world SfP dataset of ice objects is constructed, providing ground-truth surface normals and polarization observations to benchmark learning-based SfP methods in complex media.

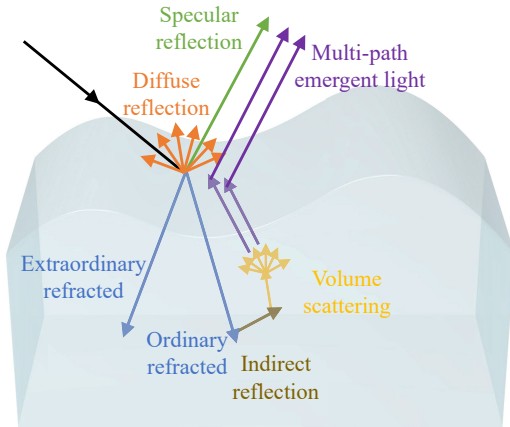

*Figure 2.* Schematic of light propagation in ice. Surface reflections and internal birefringence, internal reflections, and volume scattering collectively lead to multi-path emergent light.

## 2. Related Work

### 2.1. Ice Shape Recognition

Existing research on 3D shape perception of ice objects has primarily focused on thin ice layers or geometrically simple scenarios. Due to the high transmissivity and volumetric scattering properties of ice in the visible spectrum, early approaches generally relied on active projection-based observations, reconstructing geometry by projecting known signals onto the ice surface and analyzing the resulting responses. For example, Zuo et al. (Zuo et al., 2024) employed line laser scanning combined with precise 3D calibration to recover the shape of ice blocks, while Gou et al. (Gou et al., 2023) applied thermal pulses alongside infrared imaging to estimate ice layer thickness distributions. These methods can achieve high accuracy under controlled experimental conditions; however, their inference is heavily dependent on stable projection signals, precise calibration, and controlled imaging environments, often requiring complex and costly hardware.

In recent years, some studies have explored passive imaging paradigms for ice media. For instance, Cai et al. (Cai et al.,

2026) employed binocular stereo vision to reconstruct ice geometry, enhancing surface texture and edge information through dark-field illumination, and combining multi-view matching for geometric recovery. However, volumetric scattering and multi-path propagation in ice violate the assumptions of cross-view pixel correspondence and photometric consistency, resulting in unstable and error-prone geometric reconstruction. Even with optimized imaging setups or improved matching strategies, these issues remain largely unresolved.

Overall, existing methods often depend on additional constraints or complex hardware setups, resulting in limited robustness and scalability, and are generally applicable only to geometrically simple or well-controlled ice structures.

### 2.2. Shape from Polarization

Shape from Polarization (SfP) provides a cost-effective approach for estimating surface normals from a single viewpoint without requiring additional light sources, by analyzing the polarization state of light reflected from object surfaces. Ba et al. (Ba et al., 2020) were the first to introduce deep learning into SfP, employing neural networks to learn the complex mapping between polarization observations and both material properties and geometric structures, significantly extending the applicability of SfP to non-ideal materials.

As research has progressed, scholars have begun exploring SfP in more complex scenarios (Lei et al., 2022), such as transparent objects or underwater environments. For example, Shao et al. (Shao et al., 2023) leveraged the higher noise level in transmitted components compared to reflected components to construct a noise confidence measure and used a multi-branch network for geometric reconstruction of transparent objects. Wu et al. (Wu et al., 2025), on the other hand, exploited the advantages of polarization imaging in mitigating scattering effects, proposing an attention-based $U^2$Net framework for 3D reconstruction of underwater polarization images. However, systematic studies remain limited for materials with complex optical properties.

## 3. Proposed Method

### 3.1. Overview

To estimate the surface normals of ice media, this work proposes the IceSfP neural network, as illustrated in Fig. 3. From single-view polarization observations, three candidate surface normal maps are generated; detailed derivations are provided in Appendix A.1. Due to the complex optical propagation within the medium, these physics-based normal priors are inherently ambiguous. To address this, a structure-aware polarization consistency prior is constructed from the intensity, Degree of Linear Polarization (DoLP), and AoLP

channels of the raw polarization observations. This prior, together with the CRA module, adaptively modulates the contribution of the physics-based priors. The consistency prior and physics-based priors are jointly fed into the normal estimation branch, while features are extracted from the raw polarization observations via a separate observation branch. Finally, the network fuses these multi-source inputs to predict the surface normals of the ice medium.

### 3.2. Structure-aware Polarization Consistency Prior

In this subsection, we propose a structure-aware polarization consistency prior. It is designed to guide neural networks to focus on regions of polarization observations that exhibit spatial coherence, thereby improving the robustness of learning under complex media conditions. In ice materials, the measured polarization signals generally arise from a mixture of surface Fresnel reflection and internal light propagation effects, including volume scattering, birefringence, and anisotropic transport. When local polarization is predominantly governed by stable surface reflections, the AoLP exhibits smooth transitions between adjacent pixels, as shown in Fig. 4(b). Conversely, internal propagation effects introduce multipath interference, leading to abrupt directional shifts, phase reversals, or textural disruptions in AoLP maps. Appendix A.2 provides additional discussion on the relationship between AoLP behavior, surface normals, and volumetric scattering effects.

The autocorrelation function (ACF) is widely used in signal processing and image analysis to measure the similarity between a signal and its spatially shifted versions, making it effective for characterizing local structural regularity and stability (Chen et al., 2023; Wu et al., 2022; Yu et al., 2024; Cai et al., 2025; Shin et al., 2023). In this work, we employ ACF to measure the spatial consistency of AoLP within local neighborhoods.

To robustly handle the inherent $\pi$-periodicity of AoLP and suppress unreliable polarization measurements, we adopt a DoLP-weighted double-angle vector representation:

$$\mathbf{u}(x) = \text{DoLP}(x) \begin{bmatrix} \cos\big(2\,\text{AoLP}(x)\big) \\ \sin\big(2\,\text{AoLP}(x)\big) \end{bmatrix}. \qquad (1)$$

For a local neighborhood $\mathcal{N}_x$ centered at pixel $x$, the autocorrelation function is defined as

$$R_x(\Delta) = \sum_{\delta \in \mathcal{N}_x} \mathbf{u}(x+\delta) \cdot \mathbf{u}(x+\delta+\Delta), \qquad (2)$$

where $\Delta$ denotes a spatial displacement vector. The correlation response is normalized within each local window to reduce amplitude variations across different neighborhoods.

To summarize directional consistency, we analyze the radial decay behavior of the autocorrelation function along four

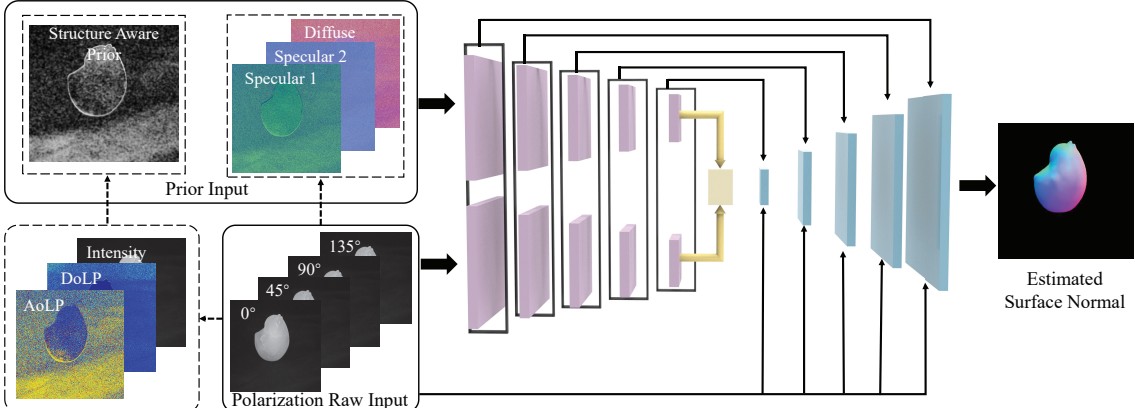

*Figure 3.* Overall architecture of the proposed method. The network adopts a dual-branch design to process raw polarization images and physics-based priors separately. Features from the two branches are fused and forwarded to the decoder via skip connections to preserve multi-scale information. Additionally, the raw polarization images are fed into the SPADE module in the decoder for spatially-adaptive feature modulation.

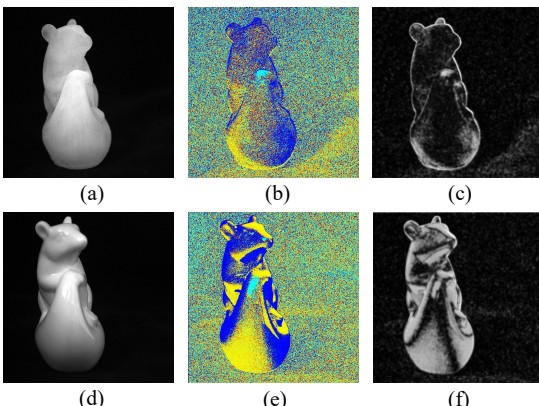

*Figure 4.* Schematic illustration of the structure-aware polarization consistency prior: (a) original image of the ice object, (b) corresponding AoLP map, and (c) consistency map; (d) original image of the ceramic object, (e) corresponding AoLP map, and (f) consistency map.

principal directions

$$\Theta = \{0°, 45°, 90°, 135°\}. \tag{3}$$

Rather than estimating a classical correlation length in the sense of stationary random processes, we define a local decay scale that characterizes how concentrated the AoLP autocorrelation is around the central pixel under discrete sampling and limited neighborhood support. Specifically, the direction-dependent correlation decay scale is defined as

$$L_x(\theta) = \min \left\{ d \, \middle| \, \frac{R_x(d\, \mathbf{e}_\theta)}{R_x(\mathbf{0})} \le e^{-1} \right\}, \tag{4}$$

As a result, $L_x$ quantifies how rapidly the autocorrelation decays from its peak, reflecting the spatial concentration of polarization coherence. A smaller $L_x$ corresponds to a highly localized and directionally coherent AoLP pattern,

which is more likely to arise in regions where polarization observations are dominated by stable surface reflections. In contrast, volumetric scattering and multipath propagation tend to disperse polarization correlations over a wider spatial support, resulting in broader autocorrelation responses and larger $L_x$.

Although local autocorrelation captures polarization consistency at the neighborhood scale, its fixed-window analysis is limited in regions with rapid structural variations, such as surface irregularities or micro-defects, which reduce local AoLP coherence. The stationary wavelet transform (SWT), a translation-invariant multi-scale signal analysis method, decomposes intensity images into multiple scales, separating low-frequency structural components from high-frequency details (Aktar et al., 2025; Hsu & Wu, 2025). By extracting horizontal, vertical, and diagonal high-frequency subbands $\{H(x), V(x), D(x)\}$, SWT captures directional structural discontinuities and complex textures. These components are aggregated into a normalized high-frequency energy measure:

$$\hat{E}(x) = \text{normalize}\big(H(x)^2 + V(x)^2 + D(x)^2\big) \in [0,1]. \tag{5}$$

The energy distribution of these subbands serves as an effective measure of the degree to which local regions deviate from the uniform surface reflection model. Areas with strong high-frequency responses indicate rapid structural changes or complex textures, which tend to reduce local polarization consistency.

Based on the above analyses, we construct a pixel-wise polarization consistency map as

$$C(x) = 1 - \big(\alpha\, \hat{L}_x + \beta\, [1 - \hat{E}(x)]\big), \tag{6}$$

where $\hat{L}_x$ is the normalized correlation decay scale, $\hat{E}(x)$ denotes the normalized SWT-based high-frequency energy. In this work, we set $\alpha + \beta = 1$ with $\alpha = 0.5$, compute the ACF over a $3 \times 3$ local window, and perform SWT analysis using a two-level Haar wavelet decomposition. The resulting map $C(x)$ (Fig. 4(c)) quantifies local polarization consistency and serves as an auxiliary prior, guiding the network toward regions where polarization observations are structurally coherent, thereby supporting more robust learning-based normal estimation in complex media.

To validate the proposed analysis, we compare two representative materials under similar geometric conditions: a ceramic object dominated by surface reflection (Fig. 4(d)) and an ice object exhibiting strong volumetric scattering (Fig. 4(a)). Due to its surface-dominated polarization behavior, the ceramic sample exhibits higher spatial coherence in AoLP, as shown in Figs. 4(b) and (e). The consistency maps in Figs. 4(c) and 4(f) indicate that the ceramic sample has significantly higher consistency than the ice sample, reflecting better alignment with a single dominant polarization mechanism. In contrast, large low-consistency regions in ice result from disruption caused by volumetric scattering and multipath effects.

### 3.3. Network Architecture and Training

This subsection details the IceSfP network, a dual-branch deep architecture that integrates raw polarization features with physics-based normal priors. The network consists of a raw polarization branch, a physics prior branch, a decoder, a CRA module, and a multi-scale feature fusion module, as illustrated in Fig. 5.

The raw polarization branch takes four-channel polarization images $(0°, 45°, 90°, 135°)$ as input and extracts multi-scale features using an EPSANet50 backbone (Zhang et al., 2023). Atrous Spatial Pyramid Pooling (ASPP) (Chen et al., 2018) is applied to enhance context awareness. The physics prior branch encodes candidate normal maps generated by the Fresnel reflection model and concatenates them with the structure-aware polarization consistency prior. Multi-scale features are then produced following the same processing pipeline as the raw polarization branch. The detailed encoder and decoder architectures are provided in Appendix A.4.

To further leverage the contribution of prior predictions at the global semantic level, a CRA module is applied at the deepest feature scale, as illustrated in Fig. 5(b). This module projects features from both modalities into a shared latent space and employs a cross-attention mechanism to selectively enhance reliable physics-based prior information while suppressing unreliable regions. The attention-weighted prior features are then fused with the raw polariza-

tion features via a residual formulation:

$$F_{\text{fused}} = F_{\text{raw}} + \gamma \cdot \text{Attention}(F_{\text{raw}}, F_{\text{physics}}), \quad (7)$$

where $\gamma$ is a learnable scaling parameter controlling the contribution of physics-guided features.

At multiple intermediate resolutions, features from the raw polarization and physics prior branches are integrated through dedicated Multi-scale Feature Fusion modules, as shown in Fig. 5(c). Each Fusion module performs channel-wise concatenation followed by lightweight convolutional operations to align feature distributions, model local structures, and adjust channel dimensions, producing scale-consistent fused representations. The fused features are forwarded to the corresponding decoder stages via skip connections, effectively preserving both polarization cues and physics-based priors across spatial scales.

The decoder progressively upsamples and fuses multi-scale features via skip connections. To mitigate information loss in deeper layers, a spatially-adaptive normalization (SPADE) module (Ba et al., 2020) is incorporated to preserve local spatial details from the polarization images. The decoder ultimately outputs a three-channel normal map.

The network is supervised using cosine similarity loss and the AoLP loss from Shao et al. (Shao et al., 2023):

$$\mathcal{L}_{\text{sim}} = \sum_{i=1}^{H} \sum_{j=1}^{W} \left( 1 - \frac{\mathbf{n}_{i,j} \cdot \hat{\mathbf{n}}_{i,j}}{\|\mathbf{n}_{i,j}\|_2 \|\hat{\mathbf{n}}_{i,j}\|_2} \right), \quad (8)$$

$$\mathcal{L}_{\text{aolp}} = \sum_{i=1}^{H} \sum_{j=1}^{W} c_{i,j} \min \left( \left| \varphi_{i,j} + \frac{\pi}{2} - \hat{\phi}_{i,j} \right|, \left| \varphi_{i,j} - \frac{\pi}{2} - \hat{\phi}_{i,j} \right| \right),$$
$$(9)$$

where $c_{i,j}$ denotes the consistency prior map, and $\varphi_{i,j}$ and $\hat{\phi}_{i,j}$ are the ground-truth and predicted azimuth angles, respectively. The overall network loss is defined as

$$\mathcal{L}_{\text{net}} = \mathcal{L}_{\text{sim}} + \lambda \mathcal{L}_{\text{aolp}}, \quad (10)$$

where the weighting factor $\lambda$ is set to 0.05 (Shao et al., 2023).

The proposed model is implemented in PyTorch and trained on eight NVIDIA Tesla A100 GPUs. The network is optimized end-to-end using the AdamW optimizer with an initial learning rate of $2 \times 10^{-4}$ and a weight decay of $1 \times 10^{-4}$ for 150 epochs, following a cosine annealing learning rate schedule. To enhance training efficiency, all images were uniformly resized to a resolution of $512 \times 512$ pixels.

## 4. Experiments

### 4.1. IceSfP Datasets

This paper constructs the IceSfP dataset, which is the first SfP dataset for ice media. It contains polarization observation data of ice samples and high-precision ground-truth

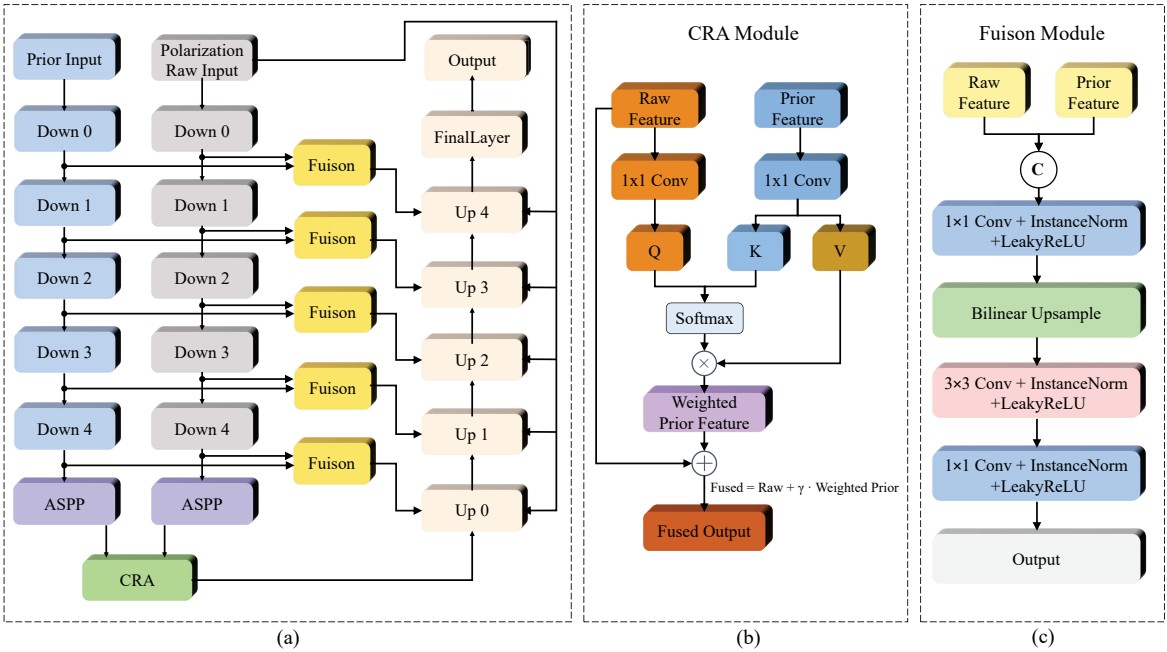

*Figure 5.* Overview of the IceSfP network. (a) Multi-branch design with a raw polarization branch and a physics prior branch guided by a structure-aware consistency prior. (b) CRA module applies cross-attention to weight physics priors and combines them with raw polarization features. (c) Multi-scale Fusion merges features at intermediate resolutions and delivers them to the decoder via skip connections.

surface normals. The ice samples were produced by pouring pure water into silicone molds cast from real objects and slowly freezing them, while the 3D geometry of the original objects was captured with a commercial desktop structured-light 3D scanner at a precision of 0.1 mm.

Polarization images were acquired under low-temperature conditions using a FLIR BFS-U3-51S5P-C polarization camera. To achieve precise geometric alignment, the captured polarization images were registered to the corresponding 3D models in MeshLab using an intensity-based mutual information method. Based on this alignment, ground-truth surface normal maps in the camera coordinate system were rendered using the Mitsuba renderer.

The IceSfP dataset comprises 16 different ice objects, each rotated from 0° to 360° in 0°–10° increments, resulting in a total of 960 samples. All polarization images were captured at the native resolution of $2448 \times 2048$ pixels. Further details of the dataset can be found in Appendix A.3.

## 4.2. Experimental Setup

To evaluate the effectiveness of the proposed method for single-view surface normal recovery in ice media, we adopt standard evaluation metrics widely used in the normal estimation literature, including the mean angular error (MAE), median angular error (MedAE), and angular accuracy under thresholds of 11.25°, 22.5°, and 30°. The angular accuracy is defined as the percentage of valid pixels whose angular error falls below a specified threshold.

Furthermore, to provide a comprehensive performance assessment, the proposed method is compared with several representative baseline approaches. Specifically, DeepSfP is a classical learning-based SfP method (Ba et al., 2020); Attention U²Net is capable of extracting stable structural features under strong scattering and complex background conditions (Wu et al., 2025); TransSfP exhibits strong adaptability in transparent and complex media (Shao et al., 2023); and SfP in the Wild (SPW) is designed for non-controlled illumination environments and demonstrates good generalization capability (Lei et al., 2022). Meanwhile, we also include the physics-based SfP method proposed by Mahmoud et al. (Mahmoud et al., 2012) as a comparison baseline.

In addition, Fig. 6 shows the analysis of the effects of the weighting parameter $\alpha$ in the proposed consistency prior. With other parameters fixed, $\alpha = 0.5$ achieves the best performance, suggesting that balancing ACF-based spatial consistency and SWT-based structural variability effectively characterizes local polarization reliability.

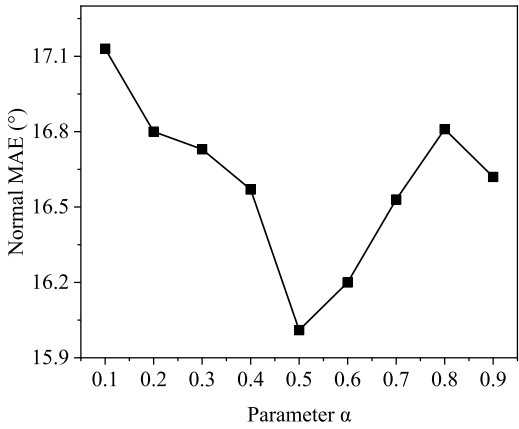

*Figure 6.* Sensitivity analysis of the weighting parameter $\alpha$ in the consistency prior.

### 4.3. Comparisons Experiment

*Table 1.* Comparison of normal estimation errors on the IceSfP dataset.

| Method | MAE ↓ | MedAE ↓ | < 11.25° ↑ | < 22.5° ↑ | < 30° ↑ |
|---|---|---|---|---|---|
| Our | **16.01°** | **13.93°** | **41.92%** | **79.58%** | **89.21%** |
| DeepSfP | 18.76° | 16.32° | 34.35% | 70.65% | 82.80% |
| Attention U$^2$Net | 19.59° | 17.62° | 29.67% | 71.17% | 83.83% |
| SPW | 20.13° | 18.34° | 25.29% | 67.27% | 82.95% |
| TransSfP | 18.75° | 16.48° | 32.13% | 73.26% | 85.01% |
| Mahmoud et al. | 62.32° | 60.09° | 0.54% | 2.75% | 4.91% |

To validate the effectiveness of the proposed method for recovering surface normals of ice media from single-view polarization images, a systematic comparison with several existing methods was conducted on the IceSfP dataset. Table 1 summarizes the average normal reconstruction results across different models. The results show that our method achieves the highest performance, yielding an average MAE improvement of 2.74° over the second-best approach.

The proposed method integrates a consistency prior with the CRA module to adaptively emphasize reliable normal cues, leading to more accurate and stable surface normal reconstruction. Purely physics-based models suffer from severe performance degradation in ice media, as complex internal optical effects violate their underlying modeling assumptions. In contrast, learning-based approaches can partially alleviate these limitations and achieve improved results. Among them, DeepSfP directly fuses polarization images with normal priors at the feature level; however, when such priors become unreliable in complex media, erroneous normal cues are easily introduced and propagated through the feature space, leading to reduced reconstruction accuracy. TransSfP reweights polarization inputs using a confidence map to suppress unreliable observations, but its confidence modeling is primarily designed to address

transmission-induced interference. As a result, it struggles to capture polarization instability caused by internal scattering and multipath propagation in complex media, limiting its effectiveness in this scenario. In addition, Attention U$^2$Net and SPW directly operate on AoLP and DoLP observations, which are often corrupted by complex internal optical effects in ice media. These distortions and inconsistencies ultimately limit the effectiveness of these methods.

*Table 2.* MAE of different methods across object models on the IceSfP dataset.

| Methods | Models (MAE ↓) | | | | | |
|---|---|---|---|---|---|---|
| | Apple | Bird | Mouse | Hemisphere | Rabbit | All |
| Our | **14.23°** | **15.06°** | **21.39°** | **9.95°** | **19.39°** | **16.01°** |
| DeepSfP | 17.55° | 15.81° | 25.42° | 11.75° | 23.24° | 18.76° |
| Attention U$^2$Net | 18.20° | 16.77° | 25.20° | 11.80° | 25.98° | 19.59° |
| SPW | 18.79° | 17.92° | 24.61° | 14.47° | 24.88° | 20.13° |
| TransSfP | 15.96° | 17.40° | 26.79° | 10.44° | 23.18° | 18.75° |
| Mahmoud et al. | 71.74° | 60.63° | 66.02° | 50.14° | 63.08° | 62.32° |

To further validate the generalization ability of the proposed method, Table 2 presents the comparison of the MAE across different object models in the IceSfP dataset. The proposed method achieves the lowest error across all object categories, indicating stable reconstruction performance across various geometries. Fig. 7 provides qualitative comparisons of different methods on various object models. Compared to existing methods, the normal maps generated by the proposed method exhibit higher overall geometric consistency. Specifically, in regions with complex internal optical effects, competing methods often fail to accurately capture the local geometric features of the object, resulting in disrupted normal distributions. In contrast, the proposed method effectively suppresses interference from these optical effects while better recovering local texture details, particularly in geometrically complex objects such as Mouse and Rabbit.

Both qualitative and quantitative results demonstrate the effectiveness and reliability of the proposed framework in ice media scenarios.

### 4.4. Generalization Across Datasets

*Table 3.* Quantitative comparison on the DeepSfP dataset.

| Method | MAE ↓ | MedAE ↓ | < 11.25° ↑ | < 22.5° ↑ | < 30° ↑ |
|---|---|---|---|---|---|
| Our | **16.33°** | **13.49°** | **47.24%** | **74.67%** | **84.65%** |
| DeepSfP | 18.60° | 15.58° | 41.11% | 67.37% | 78.85% |
| Attention U$^2$Net | 18.21° | 14.84° | 41.51% | 70.48% | 81.46% |
| SPW | 17.50° | 14.08° | 43.40% | 72.44% | 82.68% |
| TransSfP | 20.85° | 17.77° | 35.64% | 64.58% | 76.52% |
| Mahmoud et al. | 54.34° | 53.69° | 1.98% | 8.95% | 16.73% |

To further evaluate the generalization ability of the proposed

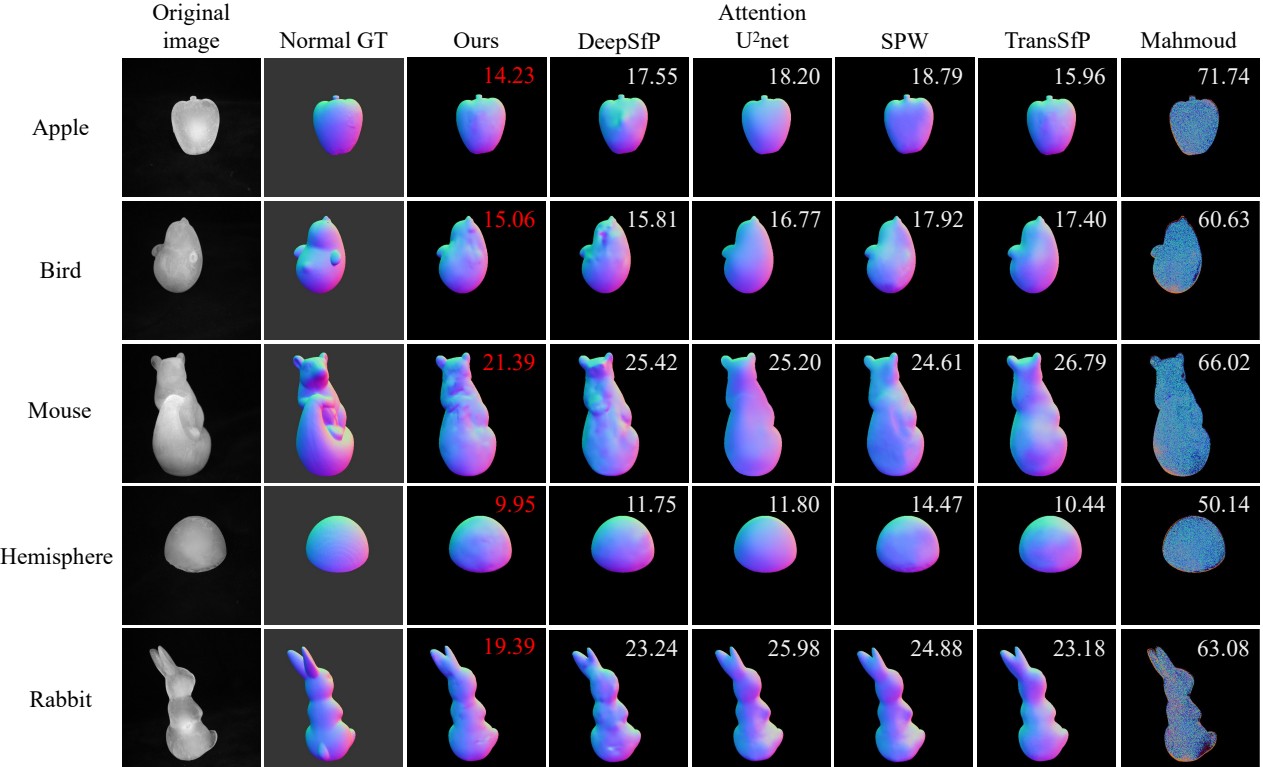

*Figure 7.* Qualitative comparison of different methods on five object models, with average MAE metrics for each model's test set.

*Table 4.* Quantitative comparison on the TransSfP dataset.

| Method | MAE ↓ | MedAE ↓ | < 11.25° ↑ | < 22.5° ↑ | < 30° ↑ |
|---|---|---|---|---|---|
| Our | **15.59°** | **12.00°** | **51.22%** | 79.37% | 87.36% |
| DeepSfP | 17.42° | 13.82° | 41.72% | 76.70% | 84.80% |
| Attention U²Net | 16.05° | 13.24° | 43.91% | **80.31%** | **87.92%** |
| SPW | 16.53° | 12.46° | 49.10% | 77.32% | 85.12% |
| TransSfP | 16.37° | 13.03° | 46.31% | 78.00% | 86.58% |
| Mahmoud et al. | 62.61° | 62.73° | 1.64% | 7.47% | 13.56% |

method beyond the IceSfP dataset, we conduct additional experiments on two external benchmarks: the DeepSfP dataset (Ba et al., 2020), which contains real-world objects with diverse materials, and the TransSfP dataset (Shao et al., 2023), which focuses on transparent objects with complex light transport effects. The training and testing splits follow the original protocols of each dataset.

Tables 3 and 4 report the quantitative results. Our method consistently achieves the best performance in terms of MAE across both datasets, outperforming the second-best method by 1.17° on DeepSfP dataset and 0.46° on TransSfP dataset. These results demonstrate that the proposed structure-aware consistency prior effectively captures the spatial reliability of polarization cues, enabling the network to suppress unreliable observations and focus on physically consistent regions. As a result, the learned model generalizes well across different materials and optical conditions.

### 4.5. Ablation Study

*Table 5.* Ablation study of different modules.

| Studied Module | MAE ↓ | MedAE ↓ | < 11.25° ↑ | < 22.5° ↑ | < 30° ↑ |
|---|---|---|---|---|---|
| W/o consistency prior | 17.84° | 15.87° | 34.84% | 72.90% | 86.13% |
| W/o CRA module | 17.44° | 15.17° | 38.48% | 73.55% | 85.90% |
| W/o SPADE module | 17.45° | 15.32° | 37.11% | 73.99% | 87.05% |
| Full | **16.01°** | **13.93°** | **41.92%** | **79.58%** | **89.21%** |

To evaluate the contributions of each module in the proposed method, we conducted ablation studies on the IceSfP dataset. Quantitative results are reported in Table 5, and corresponding qualitative comparisons are shown in Fig. 8. The full model achieves the best performance, with an average MAE of 16.01°. Removing the structure-aware polarization consistency prior increases the MAE to 17.84°, indicating that this prior guides the network to distinguish between reliable and unstable polarization regions by modeling the structural stability of AoLP, and adaptively modulates the reliability of physics-based normal priors. When the CRA module is removed, the MAE rises to 17.44°, demonstrating that this module selectively emphasizes reliable physics priors in the high-level semantic space while suppressing responses inconsistent with the raw polarization observations.

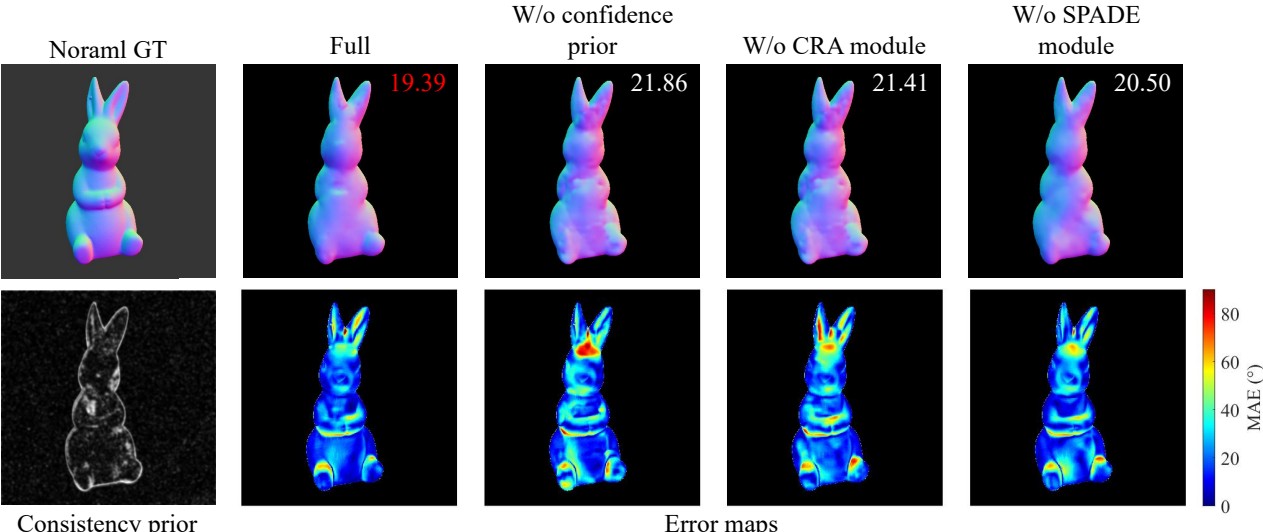

*Figure 8.* Qualitative comparison of ablation variants on the Rabbit model, with the corresponding average MAE reported.

Furthermore, removing the SPADE module increases the MAE to $17.45°$, highlighting its importance in the decoder for re-injecting local structural information from the raw polarization inputs into deep features, which is crucial for restoring fine geometric details.

Qualitative results are consistent with the quantitative trends. Removing the consistency prior or CRA module leads to unstable normal predictions in regions affected by complex optical effects, with large contiguous high-error areas in the angular error maps. Additionally, removing the SPADE module mainly results in errors concentrated along geometric boundaries, indicating reduced spatial localization in the decoded features.

Overall, the consistency prior, CRA module, and SPADE module contribute in a complementary manner by supporting reliability-guided prior utilization, effective cross-modal fusion, and spatial detail preservation, respectively.

## 5. Conclusions

This study presents the first systematic investigation of single-view SfP in ice media. To address the degradation of polarization reliability caused by complex internal light transport, we propose a structure-aware polarization consistency prior and integrate it into a dual-branch network, where the CRA module adaptively modulates the physical priors. On the IceSfP dataset, the proposed approach outperforms existing methods across different object shapes, with an average MAE of $16.01°$, demonstrating its robustness under challenging conditions such as strong scattering and multipath interference.

However, the method has only been evaluated on ice objects and mainly uses polarization consistency to adaptively adjust the physical priors. Its applicability to other types of translucent objects or highly scattering media remains to be verified. Furthermore, the framework currently assumes operation under controlled lighting conditions, which limits its performance in uncontrolled or outdoor lighting environments. Future work will extend this framework to other complex media, explicitly model the inherent uncertainties in Fresnel inversion, and further validate its performance in diverse and uncontrolled lighting scenarios.

Overall, the observation-driven physical information integration paradigm proposed in this work provides a methodologically instructive framework for high-precision geometric inference in complex media where physical models are not fully reliable.

## Acknowledgements

We gratefully thank Feng Huang (huangf@fzu.edu.cn ) and Xuesong Wang (m210210005@fzu.edu.cn ) for their valuable discussions and feedback on preliminary versions of this work.

## Impact Statement

This paper presents work aimed at advancing the fields of machine learning and computer vision. The proposed method for high-precision geometric perception of icy objects has potential applications in environmental monitoring, road icing detection, and polar research. Aside from these beneficial uses, we do not feel there are specific negative societal consequences that must be highlighted here.

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

# A. Appendix

### A.1. Physical Modeling of Ambiguous Surface Normals

This subsection presents the physical foundations of surface normal estimation and details the causes and derivation of ambiguous surface normals.

For a given polarization angle $\phi_{\text{pol}}$, the intensity at a pixel follows a sinusoidal variation under unpolarized illumination:

$$I(\phi_{\text{pol}}) = \frac{I_{\max} + I_{\min}}{2} + \frac{I_{\max} - I_{\min}}{2} \cos\big(2(\phi_{\text{pol}} - \phi)\big),$$
(11)

where $\phi$ denotes the phase angle, and $I_{\min}$ and $I_{\max}$ are the minimum and maximum observed intensities. Eq. (11) exhibits a $\pi$-ambiguity in $\phi$, since $\phi$ and $\phi + \pi$ yield the same intensity.

Based on the phase angle $\phi$, the azimuth angle $\varphi$ can be recovered as

$$\phi = \begin{cases} \varphi, & \text{if diffuse reflection dominates}, \\ \varphi - \frac{\pi}{2}, & \text{if specular reflection dominates}. \end{cases}$$
(12)

The zenith angle $\theta$ can be inferred from the degree of polarization (DoLP), defined as

$$\rho = \frac{I_{\max} - I_{\min}}{I_{\max} + I_{\min}}.$$
(13)

Under diffuse reflection dominance, the relationship between the DoLP $\rho_d$ and the zenith angle $\theta_d$ is given by

$$\rho_d = \frac{(n-1/n)^2 \sin^2 \theta_d}{2+2n^2-(n+n^{-1})^2 \sin^2 \theta_d+4\cos\theta_d\sqrt{n^2-\sin^2\theta_d}},$$
(14)

where $n$ is the refractive index of the surface. Eq. (14) can be analytically inverted to obtain $\theta_d$.

Under specular reflection dominance, the DoLP $\rho_s$ relates to the zenith angle $\theta_s$ as

$$\rho_s = \frac{2\sin^2\theta_s \cos\theta_s \sqrt{n^2 - \sin^2\theta_s}}{n^2 - \sin^2\theta_s - n^2\sin^2\theta_s + 2\sin^4\theta_s}.$$
(15)

Due to the inherent non-uniqueness of the Fresnel polarization model in Eq. 15, a single DoLP value $\rho_s$ can correspond to two feasible zenith angle solutions $(\theta_{s,1}, \theta_{s,2})$ without additional geometric or illumination constraints, leading to ambiguous surface normals.

The surface normal vector can be represented as

$$\mathbf{N} = \begin{bmatrix} \cos\phi\sin\theta \\ \sin\phi\sin\theta \\ \cos\theta \end{bmatrix}.$$
(16)

Under diffuse reflection-dominant conditions, the ambiguous surface normal is computed as

$$\mathbf{N}_d = \begin{bmatrix} \cos\phi_d\sin\theta_d \\ \sin\phi_d\sin\theta_d \\ \cos\theta_d \end{bmatrix}, \quad \phi_d = \varphi.$$
(17)

Under specular reflection-dominant conditions, Eq. (15) defines a non-injective mapping between $\rho_s$ and $\theta_s$. Consequently, a single $\rho_s$ may correspond to two valid zenith angle solutions, leading to two ambiguous surface normal estimates:

$$\mathbf{N}_{s,1} = \begin{bmatrix} \cos\phi_s\sin\theta_{s,1} \\ \sin\phi_s\sin\theta_{s,1} \\ \cos\theta_{s,1} \end{bmatrix}, \quad \mathbf{N}_{s,2} = \begin{bmatrix} \cos\phi_s\sin\theta_{s,2} \\ \sin\phi_s\sin\theta_{s,2} \\ \cos\theta_{s,2} \end{bmatrix}, \quad \phi_s = \varphi + \frac{\pi}{2}.$$
(18)

In practice, real-world surfaces often exhibit a mixture of diffuse and specular reflections. Therefore, incorporating ambiguous surface normals derived from physical models complements polarization priors, enhancing both the robustness and accuracy of SfP-based normal estimation.

### A.2. Physical Modeling of AoLP Ambiguity and Consistency

Under the ideal polarization imaging model, the AoLP is a deterministic function of the surface normal, up to a $\pi$ ambiguity:

$$\varphi = f(\mathbf{N}),$$
(19)

where $\mathbf{N}$ denotes the surface normal. For specular-dominant reflection, AoLP is orthogonal to the plane of incidence defined by the normal and viewing direction. For diffuse reflection, AoLP aligns with the projected normal on the image plane. In both cases, AoLP variations are caused solely by changes in surface geometry. Under ideal conditions, AoLP serves as a geometric prior under ideal surface-dominant conditions.

In polarization imaging, the AoLP can be directly computed from four polarization intensity measurements captured at $0°$, $45°$, $90°$, and $135°$ as

$$\varphi_{\text{obs}} = \frac{1}{2} \arctan \left( \frac{I_{45} - I_{135}}{I_0 - I_{90}} \right). \tag{20}$$

In practice, measurements in ice dielectric media contain both surface-reflection and volumetric-scattering components; the former carries physically meaningful and structurally consistent polarization cues, whereas the latter introduces weakly polarized, chaotic, and perturbing signals. This decomposition can be mathematically expressed as:

$$I_\theta = I_\theta^r + I_\theta^v, \quad \theta \in \{0°, 45°, 90°, 135°\}, \tag{21}$$

where superscripts $r$ and $v$ denote surface-reflected and volumetrically scattered components, respectively.

Substituting the above decomposition into Eq. (20), the observed AoLP can be written as

$$\varphi_{\text{obs}} = \frac{1}{2} \arctan \left( \frac{(I_{45}^r - I_{135}^r) + (I_{45}^v - I_{135}^v)}{(I_0^r - I_{90}^r) + (I_0^v - I_{90}^v)} \right). \tag{22}$$

The volumetric scattering terms, $(I_{45}^v - I_{135}^v)$ and $(I_0^v - I_{90}^v)$, exhibit spatially irregular or rapidly varying behavior due to complex internal light transport mechanisms, including subsurface scattering, birefringence, and multipath propagation. As a result, these perturbations introduce abrupt and disordered variations in Eq. 22. Moreover, the nonlinear nature of the arctangent operation further amplifies small fluctuations in the differential polarization terms, thereby inducing sudden variations in the AoLP. In contrast, the surface reflection terms, $(I_{45}^r - I_{135}^r)$ and $(I_0^r - I_{90}^r)$, are primarily governed by the Fresnel reflection mechanism, whose polarization state is determined by the local surface geometry. Consequently, these terms vary smoothly across neighboring pixels, and the corresponding AoLP preserves strong geometric consistency in the spatial domain.

Based on these physical characteristics, when the local AoLP field exhibits high spatial autocorrelation, it can be interpreted as a statistical manifestation of continuously varying surface normals in the image domain. This observation indicates that the polarization measurements in such regions are more likely to exhibit structural consistency with surface-dominant polarization models.

In summary, by quantifying the local spatial autocorrelation of AoLP, the proposed method translates physical consistency into a computable consistency measure. Furthermore, incorporating this prior into a neural network guides the learning process toward regions where polarization cues are structurally coherent, thereby supporting more robust surface normal estimation, effectively suppressing errors induced by volumetric scattering.

### A.3. Construction and Acquisition of the IceSfP Dataset

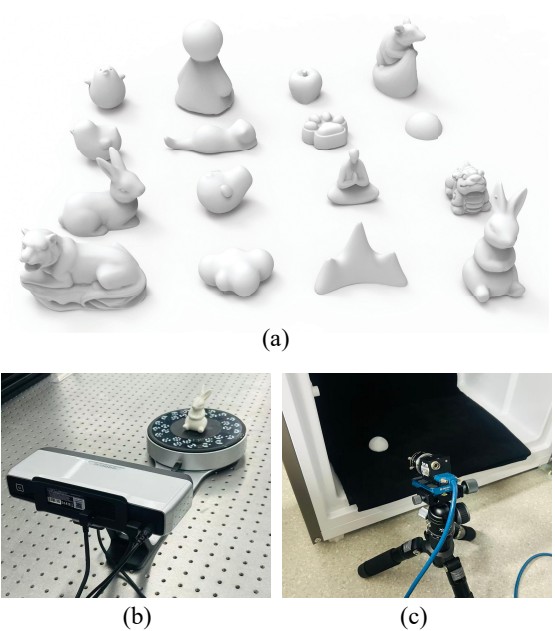

(a)

(b)          (c)

*Figure 9.* Experimental setup and data acquisition pipeline for the IceSfP dataset. (a) Illustration of the ice object models included in the IceSfP dataset. (b) Illustration of the acquisition of high-precision reference geometry of the target objects. (c) Setup for the polarization image capture under low-temperature conditions.

This appendix describes the construction process and data acquisition setup of the IceSfP dataset. To address the lack of benchmark datasets for polarization-based 3D reconstruction in ice media, we construct the IceSfP dataset, which is the first real-world dataset specifically designed for ice surface SfP. The dataset contains a diverse set of ice object geometries and provides high-precision ground-truth (GT) surface normals together with synchronously captured polarization observations.

The ice samples in the IceSfP dataset are fabricated using silicone molds cast from real objects, whose corresponding original geometries are shown in Fig. 9(a). The molds are externally reinforced with rigid shells to ensure geometric stability during the freezing process. Pure water is used to form the ice samples, which are produced via slow freezing over several hours under near-freezing conditions (approximately 0 °C). During freezing, air bubbles and micro-cracks

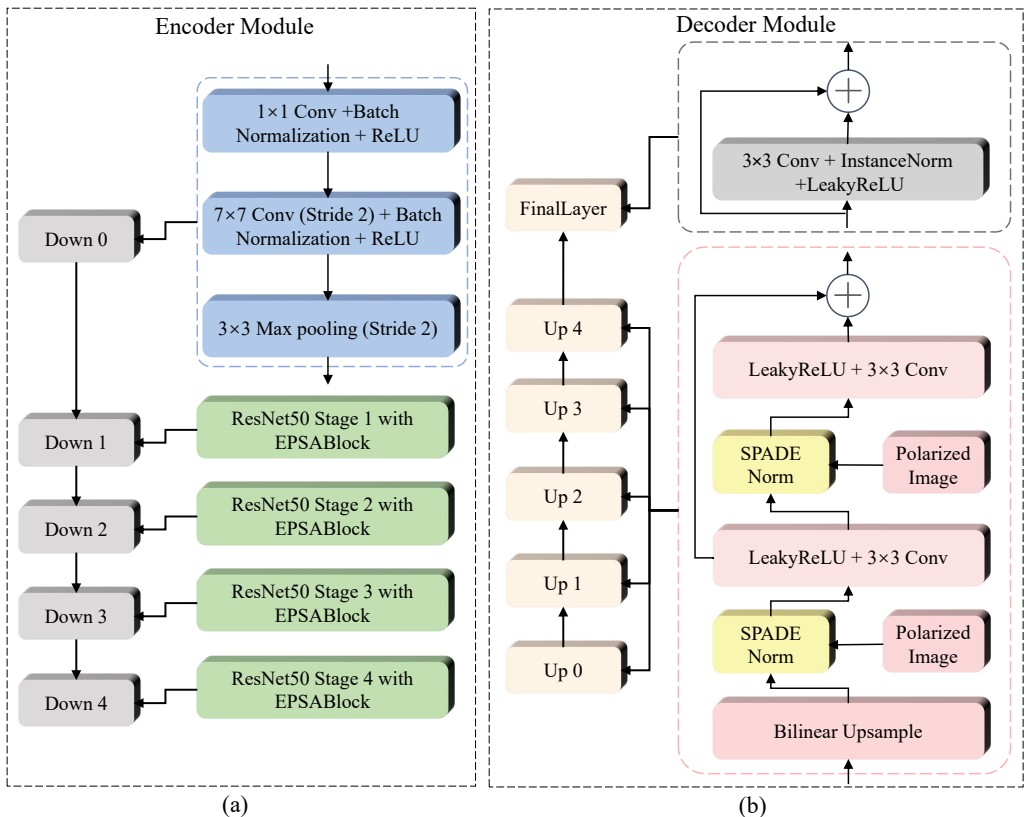

*Figure 10.* Structure of the network modules. (a) Encoder Module and (b) Decoder Module.

naturally form and are intentionally preserved to enhance the diversity and realism of the ice media, introducing internal scattering and refraction effects commonly observed in real ice.

High-precision reference geometry is obtained using a commercial structured-light 3D scanner with a nominal accuracy of 0.1 mm, as illustrated in Fig. 9(b). To generate ground-truth surface normals, the captured polarization images are aligned with their corresponding 3D object models using a mutual-information-based registration method implemented in MeshLab. The registration process consists of automatic optimization followed by manual refinement to ensure accurate global alignment. Based on the registered 3D meshes, GT surface normal maps are rendered in the camera coordinate system using the Mitsuba renderer.

The overall data acquisition pipeline is shown in Fig. 9(c). The polarization camera is mounted above the target object with a slight tilt angle, and stable indoor ambient illumination is used as a non-polarized light source. Throughout the acquisition process, the relative spatial configuration among the camera, object, and light source remains fixed. To suppress background reflections and stray light, a black diffuse cloth is placed behind the ice samples. Each acquisition produces a single snapshot polarization image containing four

polarization channels corresponding to analyzer orientations of 0°, 45°, 90°, and 135°.

For experimental evaluation, the dataset is split at the object level, with five ice objects reserved for testing and the remaining objects used for training. The test objects do not appear during training, ensuring that the evaluation objectively reflects the generalization performance of the model on previously unseen objects.

### A.4. Encoder and Decoder Architectures

This appendix details the encoder and decoder architectures, shown in Fig. 10(a) and 10(b), respectively. The encoder is based on ResNet-50, with its standard Bottleneck blocks replaced by EPSA blocks to enhance attention to spatial and channel-wise features. The decoder incorporates SPADE modules for conditional feature fusion and produces the final surface normals through the FinalLayer.

