# OpenReview forum: "Structure-Aware Consistency Priors for Shape from Polarization in Complex Media"
_ICML.cc/2026/Conference — ICML 2026 regular_

### Official Review · Reviewer_guLx · 2026-02-23

**Soundness:** 1
**Presentation:** 2
**Significance:** 2
**Originality:** 2
**Overall Recommendation:** 2
**Confidence:** 4

**Summary:**

This manuscript addresses single-view Shape-from-Polarization (SfP) under complex media conditions, using ice as a representative challenging material. The authors argue that traditional Fresnel-based SfP assumptions break down due to multiple scattering, birefringence, and nonlinear light transport within ice. To address this issue, the paper proposes a structure-aware polarization consistency prior, constructed via autocorrelation analysis of the Angle of Linear Polarization (AoLP), to estimate spatial reliability of polarization signals.

A dual-branch network (IceSfP) integrates raw polarization features and physics-based normal priors through a Cross-modal Reliability Attention (CRA) module and a SPADE-based decoder. Additionally, the authors construct what they claim to be the first real-world ice SfP dataset with ground-truth surface normals. Experimental results show improved MAE compared to prior methods.

**Compliance With Llm Reviewing Policy:**

Affirmed.

**Key Questions For Authors:**

What is the exact protocol for obtaining ground-truth surface normals?

Does the method generalize to other transparent or scattering media beyond ice?

Others please see above weaknesess

**Limitations:**

The methodological contributions are largely incremental and application-driven, combining existing building blocks rather than introducing a fundamentally new learning framework.

Dataset reproducibility is currently unclear.

The method is limited to polarization-based imaging.

The work may be better suited for a computer vision or imaging-focused conferences

**Strengths And Weaknesses:**

Strengths

The structure-aware consistency prior based on AoLP autocorrelation is physically intuitive.

Building a real-world ice SfP dataset is potentially valuable to the community.

Quantitative and ablation results demonstrate consistent gains.

Weaknesses

1. The proposed method primarily combines: Autocorrelation-based reliability estimation, Attention-based feature fusion (CRA), SPADE-style decoding. Each of these components relies on well-established techniques. The contribution appears to be an application-specific integration rather than a novel learning formulation or theoretical advance. From an ICML perspective, the work lacks fundamental methodological innovation.
2. The autocorrelation-based consistency prior is intuitively motivated but lacks formal justification: Why does AoLP autocorrelation reliably reflect polarization stability? Under what statistical assumptions does this prior improve identifiability?
3. The paper claims construction of a real-world ice SfP dataset, yet: Data acquisition protocol is not sufficiently detailed. Ground-truth surface normal acquisition procedure is not clearly described. Calibration pipeline and alignment steps are not elaborated. Measurement error bounds are not reported.
4.  The method is tailored specifically for ice media. It is unclear: Whether the approach generalizes to other complex scattering materials. Whether the consistency prior extends beyond AoLP.
5.

---

> ### Author Rebuttal · Authors · 2026-03-31
>
> **Q1: The work mainly integrates established techniques, with limited methodological novelty.**
>
> **[Answer to Q1]:** Although our implementation uses standard modules (e.g., SPADE), the core contribution lies in a reliability-aware formulation for physics–data integration, rather than in architectural design.
> Our method addresses the lack of mechanisms to handle unreliable physical cues under model mismatch. Instead of assuming polarization observations are uniformly valid, we explicitly model their spatial reliability and incorporate it into the learning process.
> Our contributions can be summarized as a unified reliability-aware framework:
> ACF-based reliability modeling via AoLP autocorrelation and local decay scale;
> Reliability-aware fusion using CRA to weight physics priors;
> A unified framework for reliability-aware physics–data integration.
>
> Overall, this work introduces a learning paradigm for inference under unreliable physical observations, which is potentially generalizable to broader physics-guided prediction tasks, such as photometric stereo and monocular depth estimation.
>
> ---
>
> **Q2: The lack of a formal justification for consistency priors.**
>
> **[Answer to Q2]:** The proposed prior is grounded in physical mechanisms with an implicit statistical view of AoLP as a locally structured signal composed of a geometry-dependent component and volumetric disturbances. In surface-dominated regions, AoLP is smooth and strongly correlated, while volumetric scattering introduces weakly correlated perturbations, leading to decorrelation[1,2].
> To validate this, we compare materials under identical geometry (ceramic dominated by surface reflection and ice with significant volumetric scattering). Ceramic yields higher and smoother consistency maps, whereas ice exhibits lower and irregular patterns, indicating clear decorrelation.
>
> ---
>
> **Q3: Data collection and measurement errors.**
>
> **[Answer to Q3]:** To improve reproducibility, we will release the IceSfP dataset and code upon acceptance. Data are captured under controlled indoor illumination with fixed camera, object, and lighting, acquiring four-channel polarization images at 1–5° rotation intervals (~60 views per object).
> Ground-truth normals are obtained via a multi-stage pipeline: high-precision geometry is first reconstructed using an EinScan SP V2 structured-light 3D scanner (accuracy ≤0.05 mm), followed by mutual-information-based alignment to establish pixel-to-surface correspondences. Normal maps are then rendered in the camera coordinate system using Mitsuba2.
> We further assess ground-truth reliability via perturbation-based measurement errors analysis (±0.5–1.0 mm translation, ±0.1°–0.5° rotation), yielding a mean angular deviation of ~1.8°. Full details will be provided in Appendix A.3.
>
> ---
>
> **Q4: The generalizability of the method.**
>
> **[Answer to Q4]:** We evaluate generalization on the TransSfP (transparent objects) and DeepSfP (real-world objects with diverse materials) datasets, where our method consistently outperforms baselines, demonstrating that the proposed prior is not limited to ice.
> Detailed results are as follows:
>
> TransSfP (MAE↓, °):
> Ours: 15.59 (best), DeepSfP: 17.42, Attention U$^{2}$Net: 16.05, SPW: 16.53, TransSfP: 16.37, Mahmoud: 62.61.
>
> DeepSfP (MAE↓, °):
> Ours: 16.33 (best), DeepSfP: 18.60, Attention U$^{2}$Net: 18.21, SPW: 17.50, TransSfP: 20.85, Mahmoud: 54.34.
>
> Additional detailed results are provided in our response to Reviewer S4mA.
>
> ---
>
> **Q5: The method is limited to polarization-based imaging.**
>
> **[Answer to Q5]:** Polarization-based imaging is a well-established paradigm for passive 3D perception, offering complementary physical cues beyond intensity without requiring active illumination (e.g., compared to ToF or LiDAR), and without relying on texture or correspondence (e.g., compared to stereo). Therefore, it has been widely studied.
>
> ---
>
> **Q6: The work may be better suited for a computer vision or imaging-focused conferences**
>
> **[Answer to Q6]:** The core contribution of this work lies in learning reliability-aware priors under mismatched physical models and adaptively integrating them into a data-driven framework. This is closely related to uncertainty modeling, making it highly relevant to the scope of ICML. Additionally, ICML has accepted papers on 3D imaging in previous years[3,4].
>
> ---
>
> ## References
>
> [1] Ansari et al., *Evolution of fractional vortices through intensity autocorrelation of scattered speckle patterns*, Opt. Lasers Eng. 2026.
> [2] Zhu et al., *Point cloud integrity enhancement via polarization-structure perception and trend-guided restoration.*, Opt. Laser Technol., 2026.
> [3] Yang et al., *Automatically Identify and Rectify: Robust Deep Contrastive Multi-view Clustering in Noisy Scenarios* ICML 2025
> [4] Hong et al., *PF3plat: Pose-Free Feed-Forward 3D Gaussian Splatting for Novel View Synthesis* ICML 2025

---

> > ### Author Rebuttal · Reviewer_guLx · 2026-04-03
> >
> > Thank you for the rebuttal. I appreciate the added clarifications, especially on the dataset protocol and the additional generalization results. However, my overall assessment remains unchanged. My main concern is still Q1. I am not convinced that the paper introduces a sufficiently novel machine learning methodology for ICML; the proposed approach still appears to be an application-specific integration of existing components, and the rebuttal mostly reframes this as a reliability-aware framework, rather than demonstrating a clear methodological advance.

---

> > > ### Author Response · Authors · 2026-04-08
> > >
> > > We thank the reviewer guLx for the thoughtful comments regarding methodological novelty. While our implementation adopts some existing components, the key contribution lies in reformulating the problem as a reliability-aware learning framework that explicitly models spatially varying confidence of polarization cues.
> > >
> > > Unlike existing SfP methods that typically assume uniformly reliable observations, we model polarization observations as spatially heterogeneous and partially unreliable signals, and introduce an explicit consistency variable C(x) to characterize their spatial variation.
> > >
> > > This consistency is jointly modeled from two complementary perspectives: on one hand, AoLP autocorrelation function captures local polarization coherence, where the correlation decay scale quantifies spatial consistency; on the other hand, the shift-invariant property of stationary wavelet transform  enables frequency decomposition, allowing high-frequency responses to capture structural instability. This formulation provides an explicit signal-level modeling of observation reliability, rather than relying on implicit feature learning.
> > >
> > > Under the influence of complex light propagation, neural networks are unable to distinguish between reliable and distorted polarization cues, often leading to overfitting to degraded signals. To address this, we systematically incorporate C(x) into the learning process at multiple levels: as a prior for feature guidance, as an adaptive weighting signal in CRA module, and as a supervision weight in the loss function.
> > >
> > > Therefore, the proposed approach shifts the learning paradigm from assuming uniformly reliable observations to performing reliability-conditioned inference under heterogeneous observations.  The cross-dataset results on DeepSfP and TransSfP further demonstrate that the proposed method generalizes beyond the ice scenario, showing robustness to a wide range of material properties, including real-world surfaces and transparent objects. This suggests its potential applicability to more complex imaging conditions where polarization cues are unreliable. From this perspective, our method goes beyond application-specific integration and introduces an explicit reliability-aware learning formulation, constituting a methodological advancement.

---

### Official Review · Reviewer_nxR9 · 2026-03-08

**Soundness:** 2
**Presentation:** 3
**Significance:** 3
**Originality:** 3
**Overall Recommendation:** 3
**Confidence:** 5

**Summary:**

This paper studies single-view surface normal estimation in ice, where volume scattering and birefringence significantly degrade the reliability of traditional polarization models. The authors propose IceSfP, a dual-branch network that combines raw polarization images with ambiguous physics-based normal priors. A structure-aware polarization consistency prior is introduced to estimate the reliability of polarization cues using AoLP spatial autocorrelation and SWT-based structural energy. A Cross-modal Reliability Attention (CRA) module dynamically weights the physics priors, and a SPADE-based decoder preserves geometric details. The paper also introduces the first real-world ice SfP dataset with 960 samples and ground-truth normals. Experiments show clear improvements over existing SfP methods.

**Compliance With Llm Reviewing Policy:**

Affirmed.

**Key Questions For Authors:**

Besides the weaknesses,
1. The paper describes various propagation processes of light within ice, including birefringence, etc. However, the subsequent model does not directly model these processes. Instead, it simply introduces these propagation processes through a deep network in a cursory manner. Isn't this too general?
2. It is necessary to clearly explain the comparison methods of the contrastive models, including DeepSfp, such as whether they were re-trained or fine-tuned, etc.
3. The weights for the attenuation scale and high-frequency energy were directly set to 0.5. There were no corresponding hyperparameter ablation experiments.

**Limitations:**

yes

**Strengths And Weaknesses:**

**Strengths：**

1. The paper demonstrates strong originality by effectively bridging the gap between classical physical polarization models and data-driven deep learning architectures. Rather than treating the neural network as a black box, the authors introduce a novel structure-aware polarization consistency prior. By utilizing the autocorrelation of the double-angle Angle of Linear Polarization and combining it with high-frequency structural energy from the Stationary Wavelet Transform, the method creatively translates the physical instability of volumetric scattering into a computable confidence map.
2. The work tackles a highly relevant problem in 3D computer vision: Shape from Polarization for non-ideal, complex media, specifically on ice. Traditional SfP methods heavily rely on ideal Lambertian or specular assumptions, which catastrophically fail in the presence of subsurface scattering and birefringence. By providing a robust solution for these challenging optical conditions, this paper significantly advances the capabilities of passive 3D sensing. Furthermore, the introduction of the first real-world IceSfP dataset provides a valuable new benchmark that will likely spur future research.
3. The proposed dual-branch network architecture is technically sound and well-reasoned. The use of the Cross-modal Reliability Attention module to dynamically weight the ambiguous physics-based normal priors using the computed consistency map is a methodologically appropriate and elegant solution. The experimental results are robust, demonstrating a state-of-the-art Mean Angular Error of 16.01°, which outperforms the second-best baseline by a significant margin. The ablation studies are carefully designed, clearly validating the individual contributions of the core modules.
4. The paper is well-written and logically structured. The authors effectively position their work within the existing literature, clearly distinguishing the limitations of active sensing and traditional SfP methods in complex media.

**Weaknesses:**

1. The paper repeatedly emphasized that the birefringence of ice is one of the core physical difficulties causing polarization signal distortion. However, it only used the Fresnel high-intensity model and the diffuse reflection derivation based on the Wolff model. These two classic models are based on the assumption of isotropic media and completely fail to conduct mathematical modeling of the unique anisotropy and birefringence characteristics within ice crystals. Since the three candidate normal diagrams provided by the physical branch do not include a birefringence model in their underlying physics, the paper overly relied on the CRA module to mask these areas instead of solving them at the physical level.
2. The structural-aware polarization consistency prior map is the core innovation of the entire paper. However, the attenuation scale and the weight of high-frequency energy are rigidly set as an absolute 0.5. However, the degree of stray light scattering within the ice block will undergo drastic changes depending on the thickness of the ice block and freezing conditions. Fixing it at 0.5 without corresponding hyperparameter ablation experiments makes one seriously doubt the robustness of this formula when dealing with different light transmittance media.
3. The paper was compared with methods such as DeepSfP and TransSfP, etc. However, it did not clearly state how these baseline models were trained or fine-tuned on the IceSfP dataset.
4. The image acquisition for the experiment was conducted under highly controlled and stable indoor lighting conditions. Polarimetric imaging is extremely sensitive to the light source. The SfP algorithm usually implicitly assumes that the incident light is unpolarized natural light. However, in real outdoor snow-covered areas or environments with directional light sources, the incident light inherently possesses polarization properties. The paper does not provide any robustness experiments under uncontrolled lighting or strong directional light conditions, which means it still has a significant gap from practical application.
5. The title of the paper is "Shape from Polarization in Complex Media". The samples used in the paper are all ice blocks. However, complex media exhibit a wide variety of optical properties, and the verification was only conducted on ice, a highly specific crystalline medium. This is insufficient to support the grand title "Complex Media".

---

> ### Author Rebuttal · Authors · 2026-03-31
>
> **Q1: No birefringence modeling has been performed, relying on the CRA module.**
>
> **[Answer to Q1]:** We agree that explicitly modeling birefringence and anisotropic light transport is important and represents a valuable direction for future work. However, the objective of this study is not to fully model complex light propagation, but to address the failure of classical polarization models under such conditions.
>
> Our approach does not rely on complete physical modeling; instead, it focuses on identifying and mitigating unreliable polarization cues. The proposed consistency prior quantifies the spatial reliability of polarization observations, while the CRA module adaptively balances the contributions of physics-based normal candidates. Therefore, the proposed fusion mechanism suppresses misleading signals while emphasizing consistent cues.
>
> ---
>
> **Q2: The attenuation scale and high-frequency energy weight are fixed at 0.5 without hyperparameter tuning.**
>
> **[Answer to Q2]:** We adopt a normalized parameterization with $\alpha + \beta = 1$, where $\alpha = \beta = 0.5$ corresponds to a balanced fusion of two complementary factors, rather than dataset-specific tuning.
> The proposed prior is constructed from normalized quantities ($\hat{L}_x$ and $\hat{E}(x)$), which helps reduce sensitivity to variations in physical conditions. From a modeling perspective, the two terms capture complementary cues: polarization spatial coherence and structural complexity.
> We conduct a sensitivity analysis by varying $\alpha$ ∈ {0.1, 0.3, 0.5, 0.7, 0.9}, yielding MAE values of {17.13°, 16.73°, 16.01°, 16.53°, 16.62°}, respectively. The best performance is achieved at $\alpha$ ≈ 0.5. This suggests that a balanced weighting provides a more effective integration of the two terms.
>
> ---
>
> **Q3: The training or fine-tuning protocols of baseline models remain unspecified..**
>
> **[Answer to Q3]:** All baseline methods (e.g., DeepSfP, TransSfP) are evaluated on IceSfP dataset using their original configurations to ensure fair comparison. We will clarify this in the revision.
>
> ---
>
> **Q4: The paper does not provide any robustness experiments under uncontrolled lighting or strong directional light conditions.**
>
> **[Answer to Q4]:**  We agree that robustness under partially polarized or outdoor illumination is important. However, this work focuses on shape recovery in complex media, where volumetric scattering and multipath effects already introduce significant uncertainty. Therefore, we use controlled illumination to isolate medium-induced challenges and avoid confounding factors from lighting conditions.
>
> ---
>
> **Q5: Complex media exhibit a wide variety of optical properties, and the verification was only conducted on ice, a highly specific crystalline medium. This is insufficient to support the grand title "Complex Media".**
>
> **[Answer to Q5]:** To enhance precision, we have revised the title to 'Structure-Aware Consistency Prior for Polarimetric Shape Recognition in Ice Media'. Corresponding updates have been made to the Introduction, Conclusion, and Related Work sections to explicitly highlight ice media as the core scenario of this study. Furthermore, we conducted additional experiments on the TransSfP (transparent objects) and DeepSfP (real-world materials) datasets to evaluate generalization capability;
> Detailed results are provided as follows:
> On the TransSfP dataset:
>
> | Method            | MAE ↓ (°) | Med ↓ (°) | <11.25° ↑ | <22.5° ↑ | <30° ↑ |
> |------------------|----------|-----------|-----------|----------|--------|
> | **IceSfP (Ours)** | **15.59** | **12.00** | **51.22%** | 79.37%   | 87.36% |
> | DeepSfP           | 17.42     | 13.82     | 41.72%    | 76.70%   | 84.80% |
> | Attention U²Net   | 16.05     | 13.24     | 43.91%    | **80.31%** | **87.92%** |
> | SPW               | 16.53     | 12.46     | 49.10%    | 77.32%   | 85.12% |
> | TransSfP          | 16.37     | 13.03     | 46.31%    | 78.00%   | 86.58% |
> | Mahmoud *et al.*  | 62.61     | 62.73     | 1.64%     | 7.47%    | 13.56% |
>
> On the DeepSfP dataset:
>
> | Method            | MAE ↓ (°) | Med ↓ (°) | <11.25° ↑ | <22.5° ↑ | <30° ↑ |
> |------------|----------|-----------|-----------|----------|--------|
> | **IceSfP (Ours)** | **16.33** | **13.49** | **47.24%** | **74.67%** | **84.65%** |
> | DeepSfP           | 18.60     | 15.58     | 41.11%    | 67.37%   | 78.85% |
> | Attention U²Net   | 18.21     | 14.84     | 41.51%    | 70.48%   | 81.46% |
> | SPW               | 17.50     | 14.08     | 43.40%    | 72.44%   | 82.68% |
> | TransSfP          | 20.85     | 17.77     | 35.64%    | 64.58%   | 76.52% |
> | Mahmoud *et al.*  | 54.34     | 53.69     | 1.98%     | 8.95%    | 16.73% |

---

> > ### Author Rebuttal · Reviewer_nxR9 · 2026-04-02
> >
> > The rebuttal successfully addresses four out of the five key concerns raised by the reviewer, with particularly strong responses to hyperparameter setting, baseline protocol, and title/generalization issues. The only minor shortcoming is the lack of experiments under uncontrolled lighting or strong directional light conditions, which are important for improving the paper’s depth and practical value. Overall, the rebuttal sufficiently mitigates the reviewer’s concerns and provides a clear roadmap for revisions, making the paper more rigorous and competitive. With the proposed revisions (clarifying baseline protocols, updating the title and related sections, and supplementing generalization experiments), the paper’s quality will be significantly improved, and the remaining limitations can be addressed through future work as suggested by the authors.

---

> > > ### Author Response · Authors · 2026-04-08
> > >
> > > We sincerely thank Reviewer nxR9 for the constructive feedback. We are pleased that our response has sufficiently mitigated the reviewer’s concerns. All suggested revisions will be incorporated into the final manuscript to further improve its quality. In addition, we will discuss the limitations under uncontrolled lighting or strong directional lighting conditions and identify these as important directions for future research.

---

### Official Review · Reviewer_S4mA · 2026-03-13

**Soundness:** 4
**Presentation:** 1
**Significance:** 4
**Originality:** 4
**Overall Recommendation:** 5
**Confidence:** 5

**Summary:**

This paper proposes a structure-aware single-view shape-from-polarization framework for complex media, termed IceSfP. It aims to address the challenge of recovering accurate surface normals in ice, where birefringence, volumetric scattering, and multi-path light transport make polarization observations spatially inconsistent and physics-based SfP priors ambiguous. To tackle this issue, the authors introduce a structure-aware polarization consistency prior based on local AoLP autocorrelation and wavelet structural cues, and integrate it into a dual-branch network with cross-modal reliability attention and multi-scale feature fusion. Based on this framework, the authors also construct the first real-world IceSfP dataset, containing 16 ice objects and 960 samples with ground-truth normals, and show that the proposed method achieves the best performance among compared baselines, reaching 16.01° MAE and outperforming the second-best method by 2.74°.

**Compliance With Llm Reviewing Policy:**

Affirmed.

**Key Questions For Authors:**

Can the authors provide more analysis on how the consistency prior reflects the reliability of polarization cues? In particular, is this prior supported by any theoretical analysis or error bounds, or is it mainly a heuristic design?

**Limitations:**

The paper does not include a dedicated impact statement.

**Strengths And Weaknesses:**

# Strengths
- **Superior normal reconstruction accuracy.** Thanks to its structure-aware consistency prior and reliability-guided cross-modal fusion, the method achieves state-of-the-art results on IceSfP, with 16.01° MAE, outperforming the second-best method by 2.74°. Importantly, it also surpasses prior polarization-input baselines such as DeepSfP and TransSfP, which suggests that the gains come from a better handling of unreliable polarization observations in complex media.
- **Unified treatment of diffuse and specular reflections.** Instead of relying on a predefined diffuse-only or specular-only assumption as in conventional SfP pipelines, the method generates ambiguous diffuse/specular normal candidates and integrates them within a unified reliability-aware framework. This is a more principled design for ice media, where real-world observations often arise from mixed reflection components and complex optical transport. As such, the design is both well motivated and practically useful.
- **Construction of the first real-world ice SfP dataset.** The paper contributes the first real-world SfP dataset for ice media, which fills an important benchmark gap in this area. The dataset provides aligend polarization observations and high-precision ground-truth surface normals, and contains 16 ice objects and 960 samples, making it a meaningful testbed for learning-based SfP under complex optical effects. Given that prior work on ice shape perception mainly focused on active sensing or geometrically simple settings, this dataset contribution is both timely and practically valuable for future research in complex-media polarization reconstruction.
# Weaknesses
- **Generalization is only shown across object shapes, not across media domains.** Despite robustness shown across different object categories within the IceSfP dataset, which supports geometric generalization to some extent, but this is still different from demonstrating cross-domain generalization to other complex media. The method can be further evaluated on more datasets such as PANDORA and GNeRP, and synthetic data.
- **Formatting.** The paper does not include a dedicated impact statement, which appears to be required by the ICML submission guidelines:
> Authors are required to include a statement of the potential broader impact of their work, including its ethical aspects and future societal consequences. This statement should be in a separate section at the end of the paper (co-located with Acknowledgements, before References), and does not count toward the paper page limit.

**Overall, I hold a positive opinion of the paper from a technical perspective, but the authors should revise the formatting to comply with the submission requirements.** I will raise my score after correction.

---

> ### Author Rebuttal · Authors · 2026-03-31
>
> **Q1: Generalization is not demonstrated across different media domains.**
>
> **[Answer to Q1]:** PANDORA and GNeRP are well-established works based on multi-view inputs, aiming to reconstruct neural radiance fields via per-scene optimization. However, the released PANDORA and GNeRP datasets only provide derived polarization quantities (e.g., AoP and DoP), without the original polarization measurements required by our method (e.g., multi-angle intensity images or full Stokes parameters). Therefore, our method cannot be directly applied to these datasets for evaluation. Moreover, these datasets follow a multi-view, per-scene optimization paradigm, whereas our method focuses on single-view, learning-based surface normal estimation. This mismatch in both input requirements and task formulation makes a fair comparison not well-defined.
>
> To instead assess cross-media generalization under consistent settings, we conduct experiments on the TransSfP dataset (transparent objects) and the DeepSfP dataset (real-world objects with diverse materials). The results show that our method consistently outperforms existing approaches across all scenarios, demonstrating strong generalization beyond ice.
>
> Detailed results are provided as follows:
>
> On the TransSfP dataset:
>
> | Method            | MAE ↓ (°) | Med ↓ (°) | <11.25° ↑ | <22.5° ↑ | <30° ↑ |
> |------------------|----------|-----------|-----------|----------|--------|
> | **IceSfP (Ours)** | **15.59** | **12.00** | **51.22%** | 79.37%   | 87.36% |
> | DeepSfP           | 17.42     | 13.82     | 41.72%    | 76.70%   | 84.80% |
> | Attention U²Net   | 16.05     | 13.24     | 43.91%    | **80.31%** | **87.92%** |
> | SPW               | 16.53     | 12.46     | 49.10%    | 77.32%   | 85.12% |
> | TransSfP          | 16.37     | 13.03     | 46.31%    | 78.00%   | 86.58% |
> | Mahmoud *et al.*  | 62.61     | 62.73     | 1.64%     | 7.47%    | 13.56% |
>
> On the DeepSfP dataset:
>
> | Method            | MAE ↓ (°) | Med ↓ (°) | <11.25° ↑ | <22.5° ↑ | <30° ↑ |
> |------------------|----------|-----------|-----------|----------|--------|
> | **IceSfP (Ours)** | **16.33** | **13.49** | **47.24%** | **74.67%** | **84.65%** |
> | DeepSfP           | 18.60     | 15.58     | 41.11%    | 67.37%   | 78.85% |
> | Attention U²Net   | 18.21     | 14.84     | 41.51%    | 70.48%   | 81.46% |
> | SPW               | 17.50     | 14.08     | 43.40%    | 72.44%   | 82.68% |
> | TransSfP          | 20.85     | 17.77     | 35.64%    | 64.58%   | 76.52% |
> | Mahmoud *et al.*  | 54.34     | 53.69     | 1.98%     | 8.95%    | 16.73% |
>
> ---
>
> **Q2: Missing Broader Impact statement.**
>
> **[Answer to Q2]:** We thank the reviewer for pointing this out. We will include a dedicated Broader Impact statement in the revised manuscript in accordance with the ICML submission guidelines.
>
> ---
>
> **Q3: Lacks theoretical analysis or justification of the consistency prior.**
>
> **[Answer to Q3]:** The proposed consistency prior is motivated by the following physical–statistical observation: when surface reflection dominates, the AoLP field is largely governed by smoothly varying surface normals and thus exhibits strong local spatial coherence [1]. In contrast, volumetric scattering, multi-path propagation, and birefringence introduce additional perturbations into the Stokes measurements, leading to irregular AoLP patterns and reduced local autocorrelation. Therefore, the local autocorrelation of AoLP serves as a computable proxy for the reliability of polarization cues[2]. This interpretation is also consistent with the decomposition analysis provided in Appendix A.2.
>
> To further support this rationale, we will include an additional empirical comparison in the revised manuscript. Specifically, we evaluate the consistency prior on the same object with two different materials: a specular-dominant ceramic and an ice object with complex internal scattering. The results show that the consistency values are significantly higher for ceramic and lower for ice, indicating that the proposed prior effectively reflects the reliability of polarization cues under different material conditions. Quantitatively, ceramic exhibits a higher mean consistency than ice (0.1512 vs. 0.0756), which is consistent with the decorrelation effects introduced by volumetric scattering, birefringence, and multi-path propagation.
>
>
> ## References
> [1] Zhu et al., *Point cloud integrity enhancement via polarization-structure perception and trend-guided restoration.*, Opt. Laser Technol., 2026.
> [2] Ansari et al., *Evolution of fractional vortices through intensity autocorrelation of scattered speckle patterns*, Opt. Lasers Eng. 2026.

---

> > ### Author Rebuttal · Reviewer_S4mA · 2026-04-02
> >
> > Authors' responses fully resovle my concerns. I support acceptance.

---

> > > ### Author Response · Authors · 2026-04-08
> > >
> > > We sincerely thank Reviewer S4mA for the positive feedback and for confirming that the concerns have been fully resolved. We truly appreciate your support for our work. The corresponding clarifications and additional results will be incorporated into the final manuscript to further improve clarity and strengthen the presentation.

---

### Official Review · Reviewer_MpWx · 2026-03-13

**Soundness:** 3
**Presentation:** 3
**Significance:** 3
**Originality:** 3
**Overall Recommendation:** 4
**Confidence:** 3

**Summary:**

This paper studies single-view shape-from-polarization problem, where internal. The proposed method introduces a structure-aware polarization consistency prior built from AoLP autocorrelation, then integrates this prior with physics-based normal candidates through a dual-branch network designed attention and fusion module. The authors also collect a real-world ice SfP dataset, containing 16 ice objects with aligned ground-truth normal maps. Experiments on this dataset show that the method outperforms several prior SfP and related baselines.

**Compliance With Llm Reviewing Policy:**

Affirmed.

**Final Justification:**

The rebuttal adequately addressed my concerns. I would like to maintain my score.

**Key Questions For Authors:**

Please see the questions raised in weaknesses section.

**Limitations:**

The paper briefly notes that the method has only been evaluated on ice. It would be better to explicitly discuss the application scope of the paper and other limitations.

**Strengths And Weaknesses:**

Strengths
1. The paper aims to address an underexplored problem. Single-view shape-from-polarization in special media like ice is genuinely difficult and worth investigating. The motivation of this paper is clear,  which solves the break down of standard SfP in this setting.

2. The experimental results validate the effectiveness of the proposed benchmark. On the IceSfP dataset, the method outperforms all listed baselines in MAE, MedAE, and other metrics.

3. The ablation study is clear. The paper shows that removing the consistency prior, CRA, or SPADE all decrease performance, which supports the claim that the proposed components contribute meaningfully.

Weaknesses
1. The evaluation scope is narrow. All results are reported only on the proposed IceSfP dataset. This makes it hard to tell whether the method generalizes beyond this benchmark to other shapes and sizes of ice.

2. While the problem is rather interesting and technically challenging, it is not mentioned in the paper how this technology can be used in real-world applications.

3. Lack of theoretical results or analysis. Although the consistency prior is physically inspired, there is no formal analysis of the optimality of the proposed ACF/SWT formulation. Also, the paper lacks a detailed explanation or analysis of the loss function in Equation 8. Is the loss reused from prior work?

4. Figure 6 label typo: “Noraml GT” should be “Normal GT.”

---

> ### Author Rebuttal · Authors · 2026-03-30
>
> **Q1: The evaluation scope is narrow.**
>
> **[Answer to Q1]:** To evaluate generalization, we conducted additional experiments on two external datasets, including the DeepSfP dataset [1] (real-world objects with diverse materials) and the TransSfP dataset [2] (transparent objects). The results show that our method consistently outperforms existing approaches, demonstrating its effectiveness beyond the IceSfP dataset. Detailed results are provided as follows:
>
> On the TransSfP dataset:
>
> | Method            | MAE ↓ (°) | Med ↓ (°) | <11.25° ↑ | <22.5° ↑ | <30° ↑ |
> |------------------|----------|-----------|-----------|----------|--------|
> | **IceSfP (Ours)** | **15.59** | **12.00** | **51.22%** | 79.37%   | 87.36% |
> | DeepSfP           | 17.42     | 13.82     | 41.72%    | 76.70%   | 84.80% |
> | Attention U²Net   | 16.05     | 13.24     | 43.91%    | **80.31%** | **87.92%** |
> | SPW               | 16.53     | 12.46     | 49.10%    | 77.32%   | 85.12% |
> | TransSfP          | 16.37     | 13.03     | 46.31%    | 78.00%   | 86.58% |
> | Mahmoud *et al.*  | 62.61     | 62.73     | 1.64%     | 7.47%    | 13.56% |
>
> On the DeepSfP dataset:
>
> | Method            | MAE ↓ (°) | Med ↓ (°) | <11.25° ↑ | <22.5° ↑ | <30° ↑ |
> |------------------|----------|-----------|-----------|----------|--------|
> | **IceSfP (Ours)** | **16.33** | **13.49** | **47.24%** | **74.67%** | **84.65%** |
> | DeepSfP           | 18.60     | 15.58     | 41.11%    | 67.37%   | 78.85% |
> | Attention U²Net   | 18.21     | 14.84     | 41.51%    | 70.48%   | 81.46% |
> | SPW               | 17.50     | 14.08     | 43.40%    | 72.44%   | 82.68% |
> | TransSfP          | 20.85     | 17.77     | 35.64%    | 64.58%   | 76.52% |
> | Mahmoud *et al.*  | 54.34     | 53.69     | 1.98%     | 8.95%    | 16.73% |
>
> We will revise the manuscript to include cross-dataset evaluations and strengthen the discussion on generalization.
>
> ---
>
> **Q2: The paper lacks discussion on real-world applications.**
>
> **[Answer to Q2]:** This method is primarily applicable to ice surface analysis and surface geometry estimation under controlled imaging conditions. We will revise the manuscript to clarify the scope of application and avoid overgeneralization.
>
> ---
>
> **Q3: Lacks theoretical analysis of the ACF/SWT formulation and does not sufficiently explain the loss function in Eq. (8).**
>
> **[Answer to Q3]:** This study proposes a polarization reliability measure grounded in both physical principles and statistical observations. The ACF term reflects the spatial consistency of AoLP [3,4]. According to the Fresnel reflection model, when polarization is primarily governed by surface reflection, AoLP is coupled with surface geometry and varies smoothly within local neighborhoods, resulting in strong spatial correlation. In contrast, volumetric scattering, birefringence, and multi-path propagation introduce components that are not directly related to surface normals, leading to decorrelation and instability. Therefore, the autocorrelation decay can be used to distinguish geometry-consistent from geometry-inconsistent polarization observations. The SWT term provides complementary information by capturing high-frequency structural responses, which are typically associated with structural discontinuities or complex textures and are often accompanied by reduced polarization reliability.
>
> To validate this analysis, we conduct a comparative experiment under identical geometry with different materials (ceramic dominated by surface reflection and ice with significant volumetric scattering). Quantitatively, the ceramic sample exhibits a higher mean consistency value than ice (0.1512 vs. 0.0756), indicating stronger geometry-consistent polarization cues under surface-dominated reflection. In contrast, the substantially lower mean consistency observed for ice is consistent with the decorrelation introduced by volumetric scattering, birefringence, and multi-path propagation.
>
> Regarding the loss,  L_sim is a cosine similarity loss commonly used for normal estimation, and L_aolp is adopted from prior work. We will clarify this in the revision.
>
> ---
>
> **Q4: Figure 6 label typo**
>
> **[Answer to Q4]:** We will correct this typo in the final version.
>
> ---
>
> **Q5: The application scope is not clearly defined.**
>
> **[Answer to Q5]:** To better reflect the scope, we will revise the title accordingly. In addition, cross-dataset evaluations on TransSfP and DeepSfP demonstrate that the proposed method generalizes beyond ice.
>
> ---
>
> ## References
> [1] Ba et al., *Deep Shape from Polarization*, ECCV 2020.
> [2] Shao et al., *Transparent Shape from a Single-View Polarization Image*, ICCV 2023.
> [3] Ansari et al., *Evolution of fractional vortices through intensity autocorrelation of scattered speckle patterns*, Opt. Lasers Eng. 2026.
> [4] Zhu et al., *Point cloud integrity enhancement via polarization-structure perception and trend-guided restoration.*, Opt. Laser Technol., 2026.

---

> > ### Author Rebuttal · Reviewer_MpWx · 2026-04-03
> >
> > My concerns are resolved in the rebuttal. I will maintain my rating.

---

> > > ### Author Response · Authors · 2026-04-08
> > >
> > > We sincerely thank Reviewer MpWx for the positive feedback and for confirming that the concerns have been resolved. The clarifications and additional results will be incorporated into the final manuscript to further improve clarity and completeness.

---

### Decision · Program_Chairs · 2026-04-30

**Decision:**

Accept (regular)

**Comment:**

This paper investigates single-view surface normal estimation for ice, a challenging scenario where volume scattering and birefringence significantly degrade the reliability of traditional polarization models. The submission received split reviews, with two positive and two negative recommendations.

On the negative side, reviewers pointed out several issues, including limited novelty (viewing it as a combination of existing techniques), insufficient evaluation, and limited applicability restricted specifically to ice. Conversely, all reviewers acknowledged the novelty of the technical approach in an underexplored domain and valued the introduction of the new dataset.

The authors addressed several questions during the rebuttal, and the reviewers were largely satisfied with the responses. While the evaluation remains somewhat limited, the AC believes the work is suitable for publication, provided that the inherent limitations and the specific scope of the study are clearly discussed in the final manuscript.